# Systematic evaluation of intratumoral and peripheral BCR repertoires in three cancers

Sofia V Krasik[1,2†], Ekaterina A Bryushkova[2,3,4†], George V Sharonov[2,3,5], Daria S Myalik[5,6], Elizaveta V Shurganova[5], Dmitry V Komarov[7], Irina A Shagina[2,3], Polina S Shpudeiko[8], Maria A Turchaninova[2,3], Maria T Vakhitova[2,3], Igor V Samoylenko[9], Dimitr T Marinov[9], Lev V Demidov[9], Vladimir E Zagaynov[5,6], Dmitriy M Chudakov[2,3,5], Ekaterina O Serebrovskaya[2,3]*

[1]Center of Life Sciences, Skolkovo Institute of Science and Technology, Moscow, Russian Federation; [2]Shemyakin-Ovchinnikov Institute of Bioorganic Chemistry RAS, Moscow, Russian Federation; [3]Institute of Translational Medicine, Pirogov Russian National Research Medical University, Moscow, Russian Federation; [4]Department of Molecular Biology, Lomonosov Moscow State University, Moscow, Russian Federation; [5]Privolzhsky Research Medical University, Nizhny Novgorod, Russian Federation; [6]Nizhny Novgorod Regional Clinical Cancer Hospital, Nizhny Novgorod, Russian Federation; [7]Volga Regional Medical Centre Under Federal Medical and Biological Agency, Nizhny Novgorod, Russian Federation; [8]Department of Bioengineering and Bioinformatics, Lomonosov Moscow State University, Moscow, Russian Federation; [9]Federal State Budgetary Institution "N.N. Blokhin National Medical Research Center of Oncology" of the Ministry of Health of Russian Federation, Moscow, Russian Federation

*For correspondence:
katyaakts@gmail.com

†These authors contributed equally to this work

Competing interest: The authors declare that no competing interests exist.

## eLife Assessment

This **useful** paper systematically evaluates B-cell receptor (BCR) repertoires across tumors, tumor-draining lymph nodes, and peripheral blood in patients with melanoma, lung adenocarcinoma, and colorectal cancer. It investigates the interplay between the tumor microenvironment and immune responses, revealing differences in BCR clonotype maturity, hypermutation, and spatial distribution. The study highlights the heterogeneity in immune responses and provides **solid** insights into the potential of tumor-infiltrating B cells for therapeutic applications, despite limitations in patient cohort size and sequencing methodology.

**Abstract** The current understanding of humoral immune response in cancer patients suggests that tumors may be infiltrated with diffuse B cells of extra-tumoral origin or may develop organized lymphoid structures, where somatic hypermutation and antigen-driven selection occur locally. These processes are believed to be significantly influenced by the tumor microenvironment through secretory factors and biased cell-cell interactions. To explore the manifestation of this influence, we used deep unbiased immunoglobulin profiling and systematically characterized the relationships between B cells in circulation, draining lymph nodes (draining LNs), and tumors in 14 patients with three human cancers. We demonstrated that draining LNs are differentially involved in the interaction with the tumor site, and that significant heterogeneity exists even between different parts of a single lymph node (LN). Next, we confirmed and elaborated upon previous observations regarding intratumoral immunoglobulin heterogeneity. We identified B cell receptor (BCR) clonotypes that were

expanded in tumors relative to draining LNs and blood and observed that these tumor-expanded clonotypes were less hypermutated than non-expanded (ubiquitous) clonotypes. Furthermore, we observed a shift in the properties of complementarity-determining region 3 of the BCR heavy chain (CDR-H3) towards less mature and less specific BCR repertoire in tumor-infiltrating B-cells compared to circulating B-cells, which may indicate less stringent control for antibody-producing B cell development in tumor microenvironment (TME). In addition, we found repertoire-level evidence that B-cells may be selected according to their CDR-H3 physicochemical properties before they activate somatic hypermutation (SHM). Altogether, our work outlines a broad picture of the differences in the tumor BCR repertoire relative to non-tumor tissues and points to the unexpected features of the SHM process.

## Introduction

B cells are a relatively undercharacterized component of the tumor microenvironment. In human tumors, infiltration with B-cells, as cells and T-cells is often a positive prognostic factor, especially in conjunction with the formation of tertiary lymphoid structures (TLSss). Mechanistically, tumor-infiltrating B-cells (TI-Bs) may act via presentation of BCR-cognate antigens to T-cells (*Zhu et al., 2015*; *Bruno et al., 2017*; *Nielsen et al., 2012*), production of pro- or anti-tumor cytokines *Somasundaram et al., 2017*; *Yang et al., 2013*; *Ammirante et al., 2010*; *Pylayeva-Gupta et al., 2016*, and production of antibodies, which may be tumor-specific *DeFalco et al., 2018* and enhance killing of tumor cells via antibody-dependent cytotoxicity (ADCC) *Kurai et al., 2007* and complement-induced cytotoxicity, enhance antigen-presentation by dendritic cells *Carmi et al., 2015* or form immune complexes that promote the activation of myeloid-derived suppressor cells (MDSC) (*Barbera-Guillem et al., 1999*). The effector functions of antibodies are highly dependent on antibody isotype *de Taeye et al., 2020* and specificity. For instance, IgG1 is the most efficient factor in ADCC, and IgG3 (together with IgG1) is capable of complement-dependent cytotoxicity (*Vidarsson et al., 2014*). These functions depend on the ability of an antibody to bind its cognate antigen, which increases during B-cell maturation and somatic hypermutation (*Clarke et al., 1985*). Isotype switching, SHM, and B-cell proliferation and differentiation are influenced by the tumor microenvironment, neoantigen burden, and presence of certain driver mutations (*Isaeva et al., 2019*). Therefore, the properties of immunoglobulins produced within the tumor are both a reflection of TME features and a factor that shapes the TME.

The whole population of immunoglobulin receptors and immunoglobulins produced within a certain tissue or cell population is known as a BCR repertoire. BCR repertoire analysis has become an important approach for characterizing adaptive immune responses in health and disease (*Cowell, 2020*; *Bradley and Thomas, 2019*; *Imkeller and Wardemann, 2018*). Starting from genomic DNA or RNA, libraries representing the sequences of variable domains of antibodies can be produced and sequenced using high throughput sequencing (*Georgiou et al., 2014*; *Boyd and Joshi, 2014*; *Turchaninova et al., 2016*; *Chaudhary and Wesemann, 2018*; *Yuzhakova et al., 2020*). Alternatively, immune receptor sequences can be derived from RNA-Seq data (*Mandric et al., 2020*; *Bolotin et al., 2017*; *Wang et al., 2022a*; *Mose et al., 2016*). Then, the properties of the variable parts of immune receptor sequences are studied using bioinformatics tools that are currently quite elaborately developed (*Foglierini et al., 2020*; *Shugay et al., 2015*; *Avram et al., 2018*; *Gupta et al., 2015*; *Gervásio et al., 2023*; *Yu et al., 2016*; *Giudicelli et al., 2017*).

For BCR repertoires, features that are considered functionally important are clonality, isotype composition, biases in V- and J-gene usage, and extent of SHM (*Kitaura et al., 2017*; *Pineda et al., 2021*; *Volpe and Kepler, 2009*; *Bashford-Rogers et al., 2019*). In addition, characteristics of the immunoglobulin CDR-H3 region, such as the number of added nucleotides, length, and amino acid physicochemical properties, have functional implications and have been linked to antibody specificity *Yu and Guan, 2014* and to the B-cell maturation stage (*Grimsholm et al., 2020*).

Correspondingly, the characteristics of the immune repertoire have been found to be associated with clinical outcomes (*Isaeva et al., 2019*; *Bolotin et al., 2017*; *Reuben et al., 2017*; *Hu et al., 2019*; *Cha et al., 2017*). Specifically, high intratumoral immunoglobulin heavy chain (IGH) expression, high IGH clonality, and a high proportion of IgG1 among all IGH transcripts were strongly correlated with higher overall survival in melanoma (*Bolotin et al., 2017*). Similarly, high IgG1 mRNA levels are positively associated with improved prognosis in early breast cancer (*Larsson et al., 2020*). A high

intratumoral IgG1 to IgA ratio was associated with improved overall survival in bladder cancer *Dyugay et al., 2022* and for *KRAS*mut but not *KRAS*wt lung adenocarcinoma (LUAD), suggesting the first link between driver mutation and B-cell response (*Isaeva et al., 2019*). In lung adenocarcinoma, breast cancer, and bladder cancer, only a subset of V-segments is associated with improved survival (*Iglesia et al., 2016*). The co-occurrence of specific antibody motifs and mutated tumor-derived peptides, presumably indicating specificity to particular tumor neoantigens, was also correlated with longer survival in colorectal cancer (*Cha et al., 2017*).

The tumor immune repertoire may be used as a source of tumor-antigen-specific antibodies for therapy development (*Zhang et al., 1995*; *Kotlan et al., 2015*; *Pavoni et al., 2007*). For instance, in melanoma 30–80% of the total BCR repertoire (*Bolotin et al., 2017*) may constitute one or several dominant B cell clones. Corresponding antibodies can be produced, verified for tumor reactivity (*Pavoni et al., 2007*), and further employed in chimeric antigen receptor T cell (CAR-T) therapy or other therapeutic approaches. The knowledge of antigen specificity of cancer-associated antibodies is currently insufficient, and it is mostly limited to autoantigens and some tumor-associated antigens (*Wu et al., 2018*; *Anderson et al., 2010*; *Yadav et al., 2019*; *Hoshino et al., 2020*; *Kunizaki et al., 2016*; *Budiu et al., 2011*).

However, even in the absence of knowledge of cognate antigens, certain hypotheses may be derived from BCR repertoire characteristics. Some individual IGHV-genes and subgroups have been associated with autoimmunity (*Matsuda et al., 1998*). Likewise, an increase in the BCR clonal expansion index (*Bashford-Rogers et al., 2013*), dominated by IgA and IgM isotypes, was associated with systemic lupus erythematosus (SLE) and Crohn's disease (*Bashford-Rogers et al., 2019*).

The ability to accurately characterize the B cell repertoire in the tumor microenvironment is of vital importance for both fundamental and clinical challenges. However, tumor tissue may not always be available for analysis; therefore, it would be beneficial to derive information about the tumor BCR repertoire from peripheral blood or draining LNs. It is not known, however, whether tumor-dominant BCR clonotypes can be found in the peripheral blood of cancer patients and at what frequencies. The ability to detect these clonotypes in a patient's peripheral blood could have a predictive value for disease prognosis. In addition, it is not known how the BCR repertoire characteristics of peripheral blood B-cells and tumor-draining LNs relate to the repertoire of TI-Bs.

Therefore, in the current study, we aimed to comprehensively and systematically evaluate tumor, lymph node, and peripheral blood BCR repertoires and their interrelationships. We show that the tumor BCR repertoire is closely related to that of draining LNs, both in clonal composition and isotype proportion. Furthermore, we observed that different LNs from one draining lymph node pool may be differentially involved in the interaction with the tumor, as reflected by the similarity of the BCR repertoire clonal composition. CDR-H3 properties indicate a less mature and less specific BCR repertoire of tumor-infiltrating B cells compared to circulating B cells. BCR clonotypes that expand in a tumor relative to other tissues are, on average, less hypermutated than non-expanded (ubiquitous) clonotypes.

## Results
### Experimental and computational study design

To systematically study the relationships between BCR repertoires in tumors and normal peripheral compartments, we performed RNA-based targeted BCR repertoire analysis from four tissue types: tumor (tum), corresponding normal tissue (norm), tumor-draining LNs, and peripheral blood mononuclear cells (PBMC), of 14 cancer patients (melanoma, n=6; lung cancer, n=4; and colorectal cancer, n=4). To account for spatial heterogeneity, we obtained three fragments of tumor tissue per patient. For draining LNs, we either dissected them into three separate pieces to study intra-LN spatial heterogeneity (lung cancer and melanoma, parts of LNs designated as LN11, LN12, LN13), or, where available, obtained three separate draining LNs (designated as LN1, LN2, LN3; *Figure 1A*). All fragments were homogenized separately into single-cell suspensions. To account for sampling noise at the level of individual cells, we obtained two replicate samples from each fragment after homogenization. As shown on *Figure 1B*, repertoires obtained from replicates at the level of cell suspension (**left panel**) show much stronger clonotype frequency correlation compared to repertoires obtained from separate tumor fragments (**right panel**). However, to determine the level of sampling noise for each

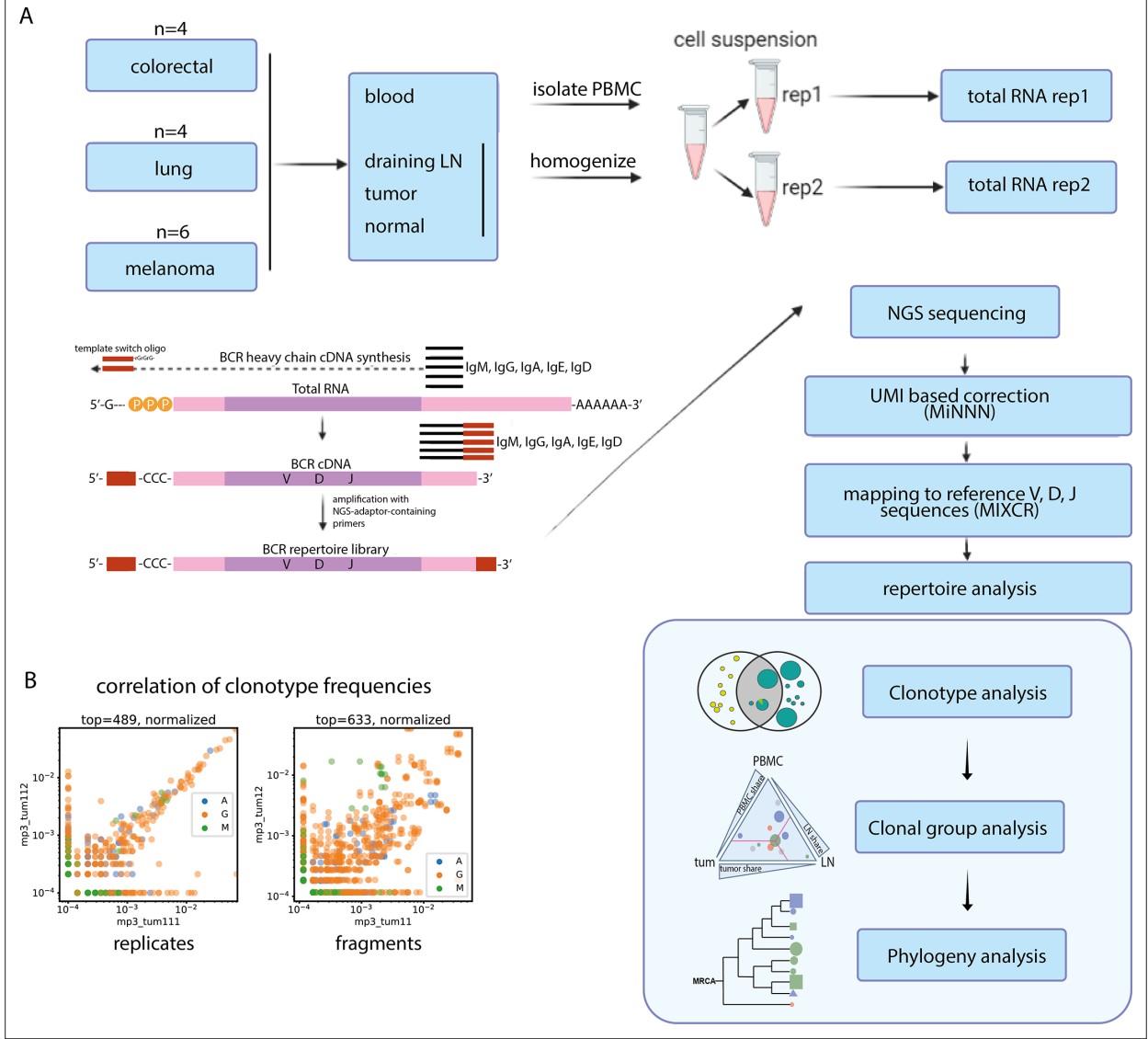

**Figure 1.** Experimental design. (**A**) Overview of the experimental design and procedures. Cell suspensions from all samples were divided into 2 replicates and processed for BCR repertoire profiling. UMI-based correction was employed to compensate for PCR-based bias. MIXCR software was used for alignment to reference BCR V, D, and J genes. The following bioinformatic analysis was performed on several levels: individual clonotype level (clonotype overlap), clonal group level (clonal group sharing between tissues), and clonal phylogeny level. (**B**) Using technical replicates at the cell suspension level allows to account for sampling bias in the clonotype frequencies.

individual clonotype, and therefore confidently identify it as significantly expanded in one sample over the other, replicates at the level of cell suspension are strictly required.

BCR repertoires libraries were obtained using the 5'-RACE (**R**apid **A**mplification of **c**DNA **E**nds) protocol as previously described *Turchaninova et al., 2016* and sequenced with 150+150 bp read length. This approach allowed us to achieve high coverage for the obtained libraries (*Supplementary file 1*) to reveal information on clonal composition, CDR-H3 properties, IgM/IgG/IgA isotypes and somatic hypermutation load within CDR-H3. For B cell clonal lineage reconstruction and phylogenetic analysis, however, 150+150 bp read length is suboptimal because it does not cover V-gene region outside CDR-H3, where hypermutations also occur. Therefore, to verify our conclusions based on the data obtained by 150+150 bp sequencing ('short repertoires'), for some of our samples we also generated BCR libraries by IG RNA Multiplex protocol (see Materials and ethods) and sequenced them at 250+250 bp read length ('long repertoires'). Libraries obtained by this protocol cover V gene sequence starting from CDR-H1 and capture most of the hypermutations in the V gene. Conclusions

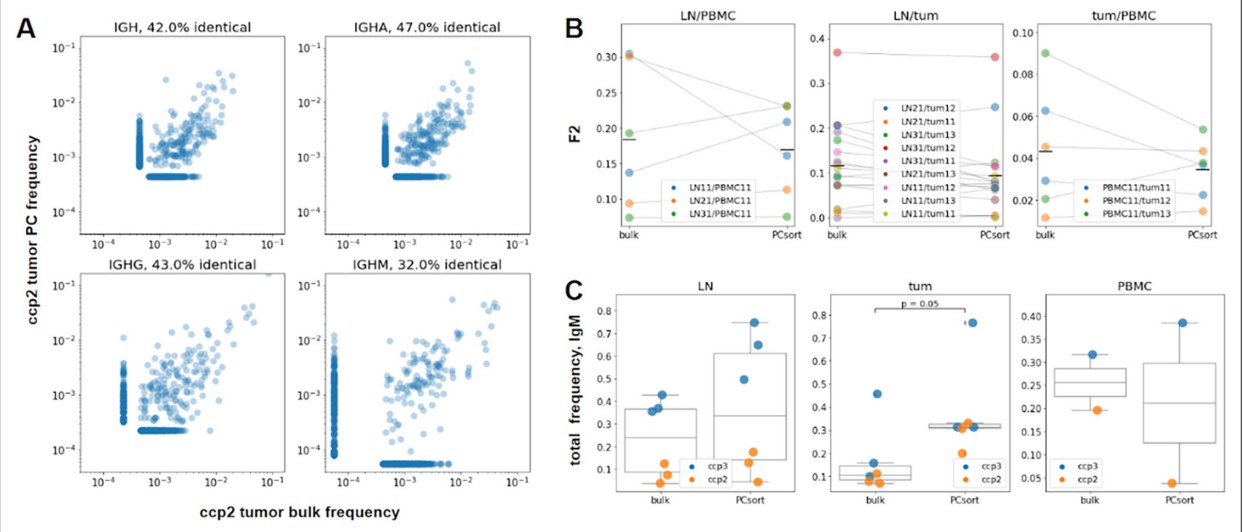

**Figure 2.** PC representation in bulk samples and repertoire similarities clonotype level, individual patient data from colorectal cancer patient ccp2 (panel **A**, **B**) or pooled patient data from colorectal cancer patients ccp2 and ccp3 (panel **C**). (**A**) Correlation plots, comparing PC and bulk frequencies of top 400 clonotypes in total tumor repertoire and isotypes A, G and M separately for colon cancer patient ccp2. Up to 47% of identical amino acid sequences were found. (**B**) Comparison of F2-overlaps for three tissue repertoires' pairs (LN and PBMC, LN and tumor, tumor and PBMC). No significant difference was found between bulk and PC-sorted samples (Wilcoxon signed-rank test). (**C**) Comparison between bulk and PC IgM proportions in lymph node, tumor and PBMC samples, the share of IgM in tumor was significantly higher in PC samples (p=0.05, Mann-Whitney test).

about clonal lineage phylogeny were drawn only when they were corroborated by 'long repertoire' analysis.

For BCR repertoire reconstruction from sequencing data, we first performed unique molecular identifier (UMI) extraction and error correction (UMI threshold = 3 for 5`RACE and 4 for Ig Multiplex libraries). Then, we used MIXCR (*Bolotin et al., 2015*) software to assemble reads into clonotypes, determine germline V, D, and J genes, isotypes, and find the boundaries of target regions, such as CDR-H3. Only UMI counts, and not read counts, were used for quantitative analysis. Clonotypes derived from only one UMI were excluded from the analysis of individual clonotype features but were used to analyze clonal lineages and hypermutation phylogeny, where sample size was crucial. Samples with 50 or less clonotypes left after preprocessing were excluded from the analysis.

## Bulk BCR repertoires are dominated by clonotypes derived from plasma cells

Previously, correlation between BCR repertoire parameters and clinical outcomes in cancer patients was mostly described using bulk RNA-Seq data (*Bolotin et al., 2017*; *Dyugay et al., 2022*). BCR repertoires extracted from RNA-Seq data represent the most dominant BCR clonotypes in the sample. These clonotypes are expected to come from antibody-secreting plasma and plasmablast cells, which are known to have as much as 500-fold higher IGH expression compared to naive and memory cells (*Tumeh et al., 2014*). Therefore, it may be assumed that as we look at bulk tumor BCR repertoire properties and their correlation with clinical outcomes, we mostly see the influence of antibody-producing plasma cells. To directly address this question, we compared RNA-based BCR repertoires obtained from sorted plasma cells/plasmablasts and bulk cell suspensions in parallel from the same samples of PBMC, LN, and tumors from patients with **ccp2** and **ccp3**. We found that the repertoire overlap between tissues, isotype composition, and clonal distribution found in BCR repertoires from bulk (unsorted) samples closely resembled those of sorted plasma cells (*Figure 2A, B and C*). One notable exception was the IgM proportion in tumors, which was significantly higher in sorted plasma cells than in unsorted TIL-B (*Figure 2B*). This indicates a significant underrepresentation of IgM-producing plasma cells in bulk tumor BCR repertoires compared to other tissues, which in turn may indicate a lower expression of BCRs in tumor-infiltrating IgM +plasma cells.

In the following experiments, we used bulk RNA-based BCR profiling with the understanding that the dominant clonotypes and B cell lineages reproducibly represented in biological replicates reflect

the presence of clonally expanded plasma cells, plasmablasts, and the most expanded memory B cell clones. In other words, the RNA-based approach mainly revealed the repertoire of the most functionally active B-cell lineages.

## Tumor/non-tumor repertoire overlap and isotype composition

First, we characterized the relative similarity of IGH repertoires derived from tumors, tumor-draining LNs, and PBMC on the individual CDR-H3 clonotype level. We define clonotype as an instance with an identical CDR-H3 nucleotide sequence and identical V- and J- segment attribution (isotype attribution may be different). Unlike other authors, here we do not pool together similar CDR-H3 sequences to account for hypermutation. (Hypermutation analysis is done separately and defined as clonal group analysis.)

As overlap metrics are dependent on overall repertoire richness, we normalized the comparison using the same number of top most frequent clonotypes of each isotype from each sample (N=109). Repertoire data for each sample were split according to the immunoglobulin isotype, and the F2 metric was calculated for each isotype separately and plotted as an individual point. We used the repertoire overlap metric F2 ($\mathbb{C}$lonotype-wise sum of geometric mean frequencies of overlapping clonotypes), which accounts for both the number and frequency of overlapping clonotypes (*Figure 3A*). As expected, significantly lower overlaps were observed between the IGH repertoires of peripheral blood and tumors compared to LN/tumor overlaps. The LN/PBMC overlap also tended to be lower, but the difference was not statistically significant. We also analyzed D metric (**Fig. S1A, D metric**), which represents the relative overlap diversity uninfluenced by clonotype frequency ($D_{ij} = d_{ij}/(d_i*d_j)$, where $d_{ij}$ is the number of clonotypes present in both samples, while $d_i$ and $d_j$ are the diversities of samples i and j, respectively). The results for D metric indicate a similar trend to that of F2 metric. This observation allows us to conclude that tumor IGH repertoires are more similar to the repertoires of tumor-draining LNs than to those of peripheral blood, both if clonotype frequency is taken into account, and when it is not.

These results are corroborated by visualization at the individual patient level, using Cytoscape network visualization platform to visualize the structure of the repertoire overlap. As exemplified by melanoma patient mp3 (*Figure 3B*), the repertoire from the tumor is closely related to the tumor-draining LN repertoire, whereas the PBMC repertoire has very few overlapping clonotypes with both tumors and draining LNs. Overall, these analyses revealed that the extent of clonal exchange between tumors and PBMC was significantly lower than that between tumors and draining LNs. The frequencies of overlapping clonotypes were also more strongly correlated between tumors and draining LNs than between tumors and peripheral blood (**Fig. S1A, R metric**). The level of clonal exchange between tissues was dependent on isotype (*Figure 3E and F*). LN/tumor overlap was higher in the IgG repertoire (*Figure 3E*), whereas PBMC/tumor overlap was lower in the IgG repertoire (*Figure 3F*) than in the IgA repertoire. This suggests that tumor-infiltrating IgG-expressing B-cells (IgG-TI-Bs) avoid systemic circulation, whereas IgA-expressing tumor-infiltrating B-cells (IgA-TI-Bs) may be found in the peripheral blood with a higher probability. In addition, the tumor repertoire was significantly more clonal (clonality calculated as [1-normalized Shannon-Wiener index] *Tumeh et al., 2014*) than the PBMC or draining LN repertoire, both overall (*Figure 3C*) and separately for the IGHM repertoire (*Figure 3D*).

We did not find a statistically significant difference in isotype composition between cancers in any of the studied tissues, with the exception of IgM percentage in melanoma tumors. In BCR repertoires from melanoma tumors, the total percentage of repertoire consisting of IgM clonotypes was significantly lower than that in other types of cancers (colorectal and lung; **Fig. S1B**). The isotype composition of non-tumor tissues correlated with the isotype composition of the tumor; this effect was less prominent in peripheral blood (*R*=0.42, p=0.028; *Figure 3G*) and more prominent for LN (*R*=0.74, p<0.01; *Figure 3H*).

## CDR-H3 properties

Analysis of averaged CDR-H3 repertoire characteristics revealed increased CDR-H3 length in tumors compared to PBMC for the total repertoire (*Figure 4A*) and also in IgA and IgG repertoires separately (**Fig. S2A, B**), but this was not the case for IgM (**Fig. S2C**). In addition, the increase in CDR-H3 length in IgA repertoires from tumor-draining LNs compared to PBMC was statistically significant (**Fig. S2A**).

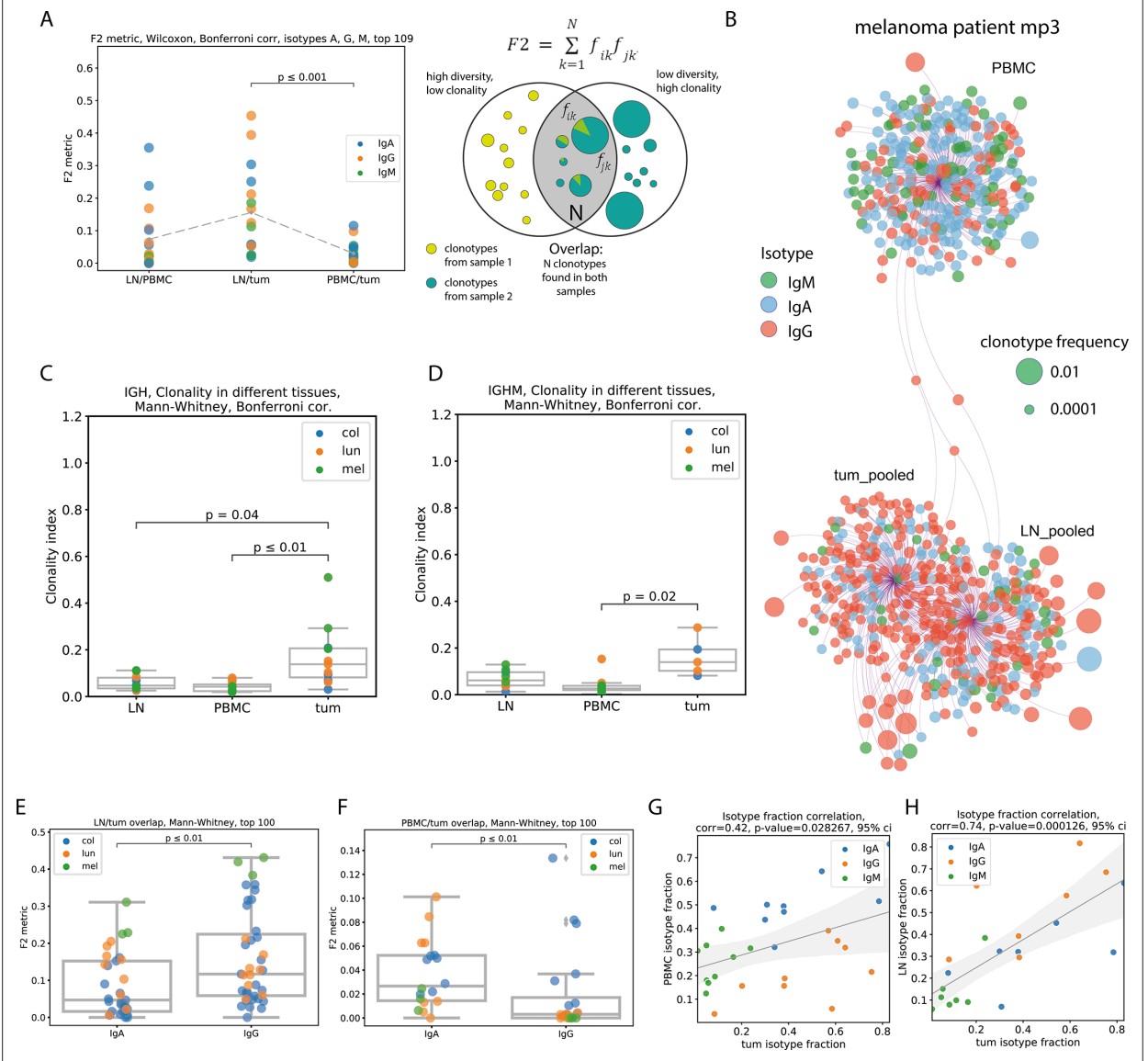

**Figure 3.** Repertoire overlap, clonality and isotype composition (clonotype level, pooled patient data (except B, data from melanoma patient 3)). (**A**) - repertoire overlap between pairs of tissues, by F2 metric, repertoires split by immunoglobulin isotype, (N=6); (**B**) network representation of Ig repertoires from PBMC, tumor-draining LN, and tumor of mp3 patient (melanoma); individual clonotypes of the same origin (PBMC, tumors, or draining LN) are shown as bubbles connected with the edges to one anchor node. Clonotypes shared between tissues were connected with two or three edges to the corresponding anchor nodes and were located between them. The size of the bubbles represents the relative frequency of clonotypes within a sample, and the color represents the isotype. The relative distance between anchor nodes corresponded to the similarity of repertoires (the number of shared clonotypes). (**C, D**) Clonality of Ig repertoires in PBMC, draining LNs, and tumors of 14 cancer patients. This reflects the presence of clonal expansion. Calculated as in *Tumeh et al., 2014*: [1-normalized Shannon-Wiener index]; (**C**) total IG repertoire; (**D**) IgM repertoire; (**E**) LN/tumor overlap for IgA and IgG repertoires (N=7); (**F**) PBMC/tumor overlap for IgA and IgG repertoires (N=9); (**G, H**) isotype fraction correlation between PBMC and tumor repertoires (G, N=9), or between LN and tumor repertoires (H, N=7).

The online version of this article includes the following figure supplement(s) for figure 3:

**Figure supplement 1.** Repertoire similarity by R metric, repertoire overlap by D metric (**A**), isotype composition by tissue type and cancer (**B**).

Interestingly, the only significant difference we found when comparing CDR-H3 lengths between cancer types was reduced IgA CDR-H3 length in colorectal cancer, especially compared to melanoma (p=0.02; *Figure 4B*). This could reflect a generally more mature IgA repertoire in colon tissues owing to the previous history of interactions with microbiota (*Benckert et al., 2011*; *Hapfelmeier et al.,*

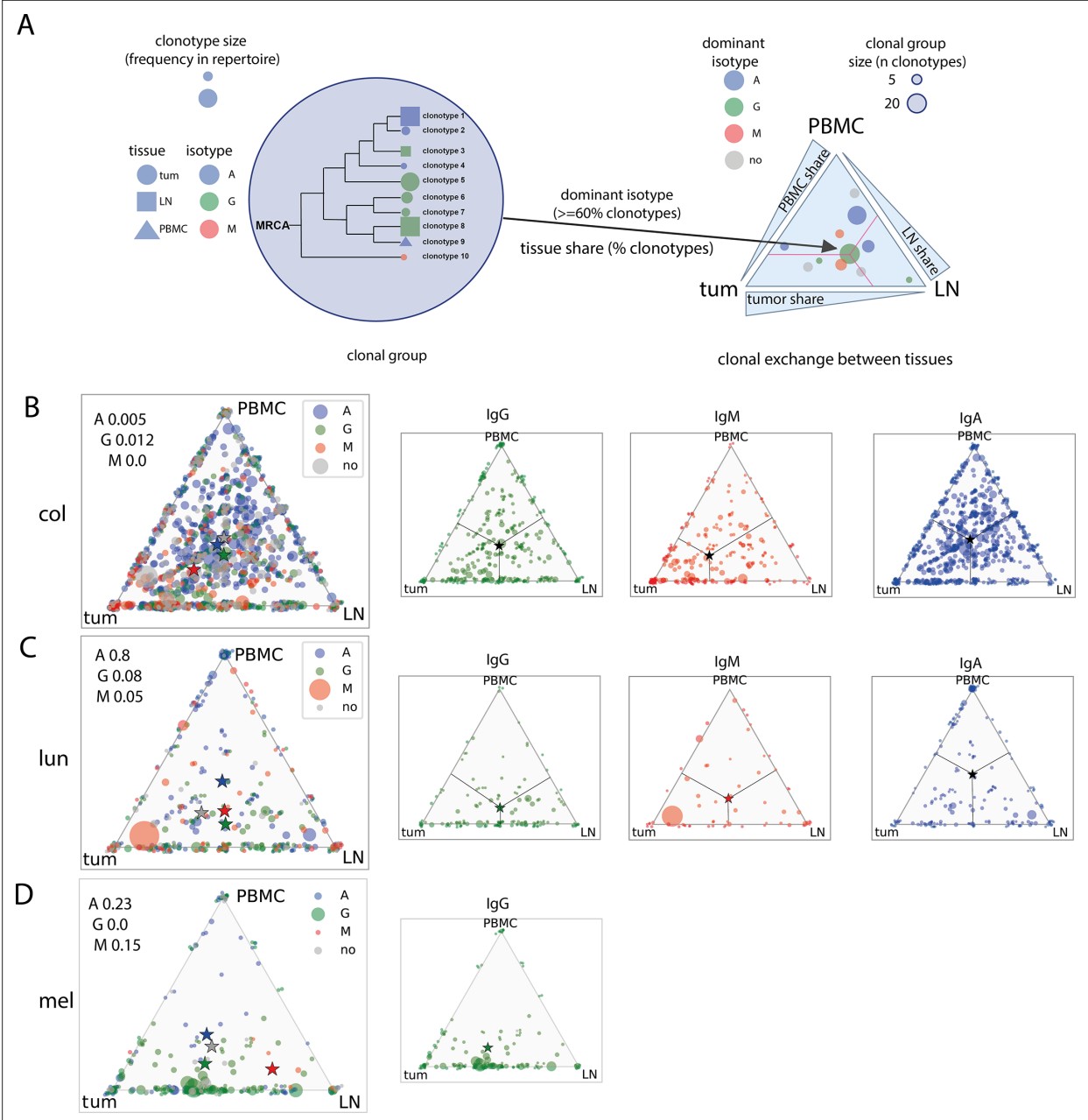

**Figure 4.** CDR-H3 amino acid properties (clonotype level, pooled patient data). (**A**) Mean amino CDR-H3 length of top 100 most frequent clonotypes from tumor, lymph node and PBMC tissues irrespective of isotype CDR-H3 are on average significantly longer in tumor than PBMC for total repertoire (N=14, p<0.01, two-sided t-test, Bonferroni correction); (**B**) Comparison of mean amino acid CDR-H3 length of 100 most frequent clonotypes for colon, lung and melanoma cancer samples from, tumor. CDR-H3s of tumor-infiltrating clonotypes were shorter for colorectal cancer patients compared to melanoma in IgA repertoires (N=11, p=0.02); (**C**) Comparison of amino acid properties in the central region of CDR-H3, for total repertoire (C-left) or IGHG repertoire (C-right), all cancers (significantly increased - red, significantly decreased - blue) two-sided t-test, Bonferroni-Holm correction; (**D**) Comparison of amino acid properties in the central region of CDR-H3, for IGHM repertoire, lung cancer (significantly increased - red, significantly decreased - blue); (p<0.01, two-sided t-test, Bonferroni-Holm correction); (**E**) Average number of mutations relative to germline for tumor samples from different types of cancers, N=14.

The online version of this article includes the following figure supplement(s) for figure 4:

**Figure supplement 1.** Mean CDR3 length for IGHA, IGHG and IGHM repertoires from lymph node, PBMC and tumor.

*2010*). However, the relationship between this difference and tumor antigen specificity remains to be verified.

To explore CDR-H3 physicochemical properties, we calculated the mean charge, hydropathy, predicted interaction strength, and Kidera factors 1–9 (kf1-kf9) for five central amino acids of the CDR-H3 region for the 100 most frequent clonotypes of each sample using VDJtools. Kidera factors are a set of scores which quantify physicochemical properties of protein sequences (*Nakai et al., 1988*). 188 physical properties of the 20 amino acids are encoded using dimension reduction techniques, to yield 9 factors which are used to quantitatively characterize physicochemical properties of amino acid sequences. Comparing between tissues, we found that kf4 value for the tumor repertoire was decreased compared to PBMC in the total repertoire, and kf5 was decreased in tumor vs. PBMC in the IgG repertoire (*Figure 4C*). In addition, kf6 value was decreased in the LN repertoire compared to PBMC. Kf4 inversely correlates with hydrophobicity, indicating a higher proportion of hydrophobic residues in BCR CDR-H3s from the tumor repertoire. Kf5 reflects a double-bend preference and has not been previously found to be significant in the context of antibody properties. Kf6 is a measure of partial specific volume; therefore, a lower kf6 value indicates less bulky amino acid residues in CDR-H3s from the LN repertoire. Between the tumor and normal tissue repertoires, the hydropathy value was lower in normal lung tissue, also indicating a higher proportion of hydrophobic residues in tumor-derived CDR-H3 repertoires (*Figure 4D*). According to *Grimsholm et al., 2020*, more mature B-cell subpopulations have higher mjenergy, disorder, kf4, kf6, and kf7 and lower CDR-H3 length, strength, volume, kf2, and kf3. Again, the mean CDR-H3 charge was negatively associated with specificity *Rabia et al., 2018* and beta-sheet propensity was associated with antibody promiscuity (*Laffy et al., 2017*). Poly-reactive and self-reactive antibodies have, on average, longer CDR-H3s *Prigent, 2016* with a higher charge *Rabia et al., 2018* and net hydrophobicity (*Wardemann et al., 2003*; *Larimore et al., 2012*; *Lecerf et al., 2019*).

Therefore, collectively, our observations suggest a less mature and less specific BCR repertoire of tumor-infiltrating B cells compared to circulating B cells and B cells infiltrating normal tissue, which may indicate less stringent control for antibody-producing B cell development in the TME.

## Immunoglobulin hypermutation analysis across tissues and isotypes

Intensity of somatic hypermutation (average number of mutations relative to the most recent common ancestor, MRCA) reflects the average extent of antigenic selection experience of the clonotypes found in a given tissue or cancer type. No significant difference was found between PBMC, tumor-draining LNs, and tumors for the top 100 most frequent clonotypes in the total repertoire as well as for the top 100 most frequent clonotypes of each isotype separately (*not shown*). This indicates that in the RNA-based BCR repertoires, in all studied tissues, the most dominant immunoglobulin clonotypes belong to cell populations with an equivalent history of antigen exposure, selection, and maturation. However, there was a statistically significant difference in the number of hypermutations in IgG between the cancers (*Figure 4E*). IGHG clonotypes from lung cancer samples show higher number of hypermutations, possibly reflecting high mutational load found in lung cancer tissue. For melanoma, another cancer known for high mutational load, no statistically significant difference was found. This may be due to higher variance between melanoma samples, which hinders the analysis, or due to the small sample size.

## Clonal exchange between tissues at the level of B cell lineages

Next, we investigated clonal exchange between the PBMCs, tumor-draining LNs, and tumors at the level of hypermutating clonal lineages, which are likely to be involved in recent and ongoing immune responses. The results are shown in *Figure 5A*. After pooling clonotype data patient-wise, clonal groups were assembled by sequence similarity, and then IGH clonotypes within a group were arranged into clonal lineages that shared a common ancestor *Barak et al., 2008* and represented a B-cell clone undergoing the affinity maturation process. Each clonotype within a clonal group was attributed to the tissue of origin, tumor, LN, and/or PBMC, and to a particular isotype. For each clonal group, the percentage of clonotypes belonging to each isotype and tissue was calculated. Clonal groups from all patients with a given cancer type were plotted on a triangle plot using the percentage of clonotypes from the tumor, LN, and PBMC as coordinates and colored according to the dominant isotype (>60%).

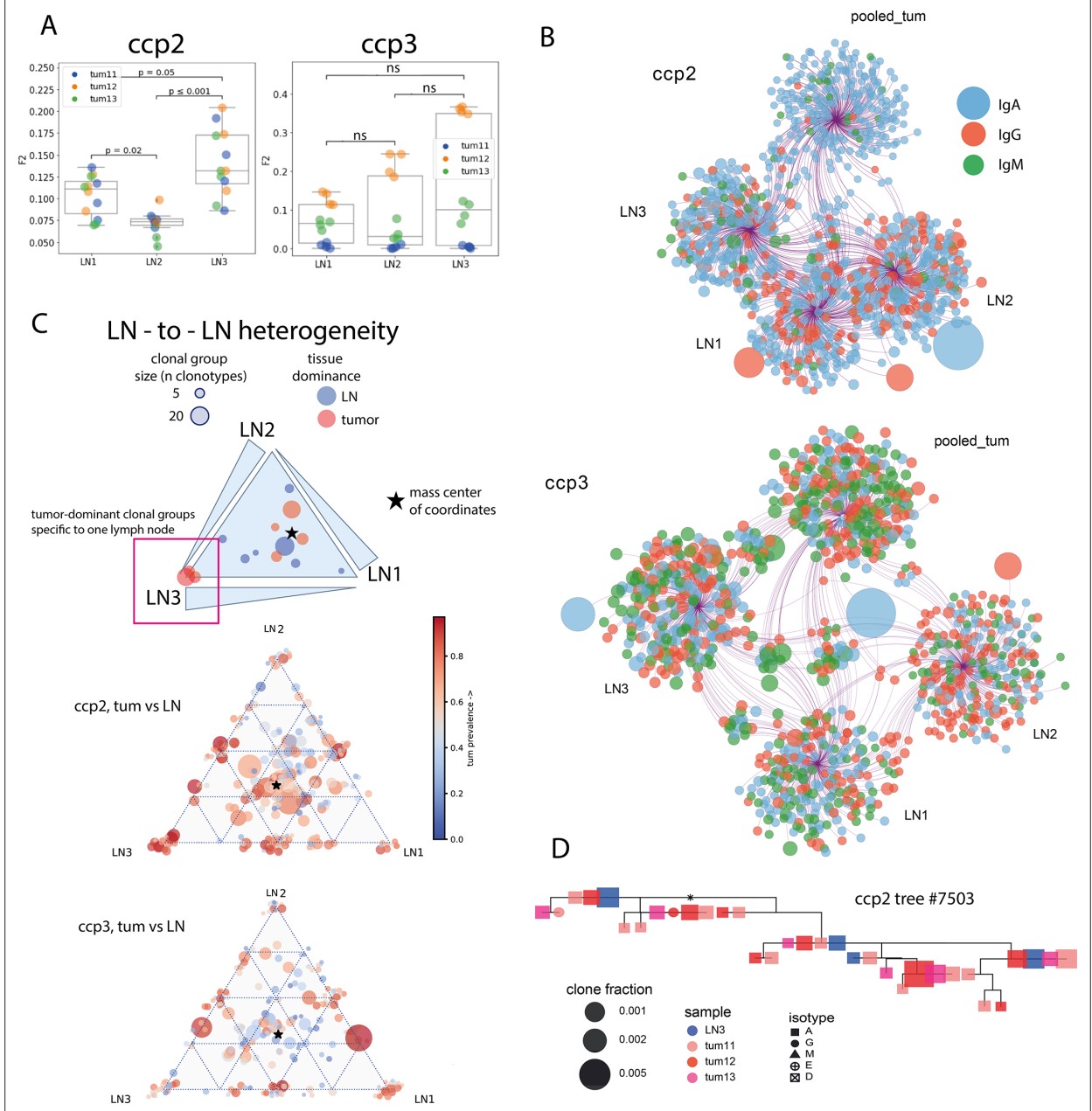

**Figure 5.** Immunoglobulin hypermutation analysis across tissues and isotypes (clonal group level, pooled patient data). (**A**) - Schematic representation of analysis and triangle plot visualization of clonal group distribution between tissues; (**B, C, D**) clonal group distribution between tissues for colorectal (**B**), lung (**C**), and melanoma (**D**); stars represent non-weighted by size mean center of triangle coordinates. Chi-2 test for goodness of fit was used to test whether each tissue equally contributed to clonal group formation.

In all studied cancer types, IgA-dominated clonal groups were evenly distributed among the three tissues, indicating no preference for IgA-switched cells towards lymphoid or tumor tissue residence. IgG-dominated clonal groups showed a preference for tumor and LN residence in lung cancer and melanoma, in accordance with the idea of tight interaction between LN and tumor in mounting anti-tumor immune responses. IgM-dominated clonal groups showed a strong preference for colorectal cancer tumors, indicating intensive intratumoral somatic hypermutation without isotype switching (*Figure 5B*, *red triangle*).

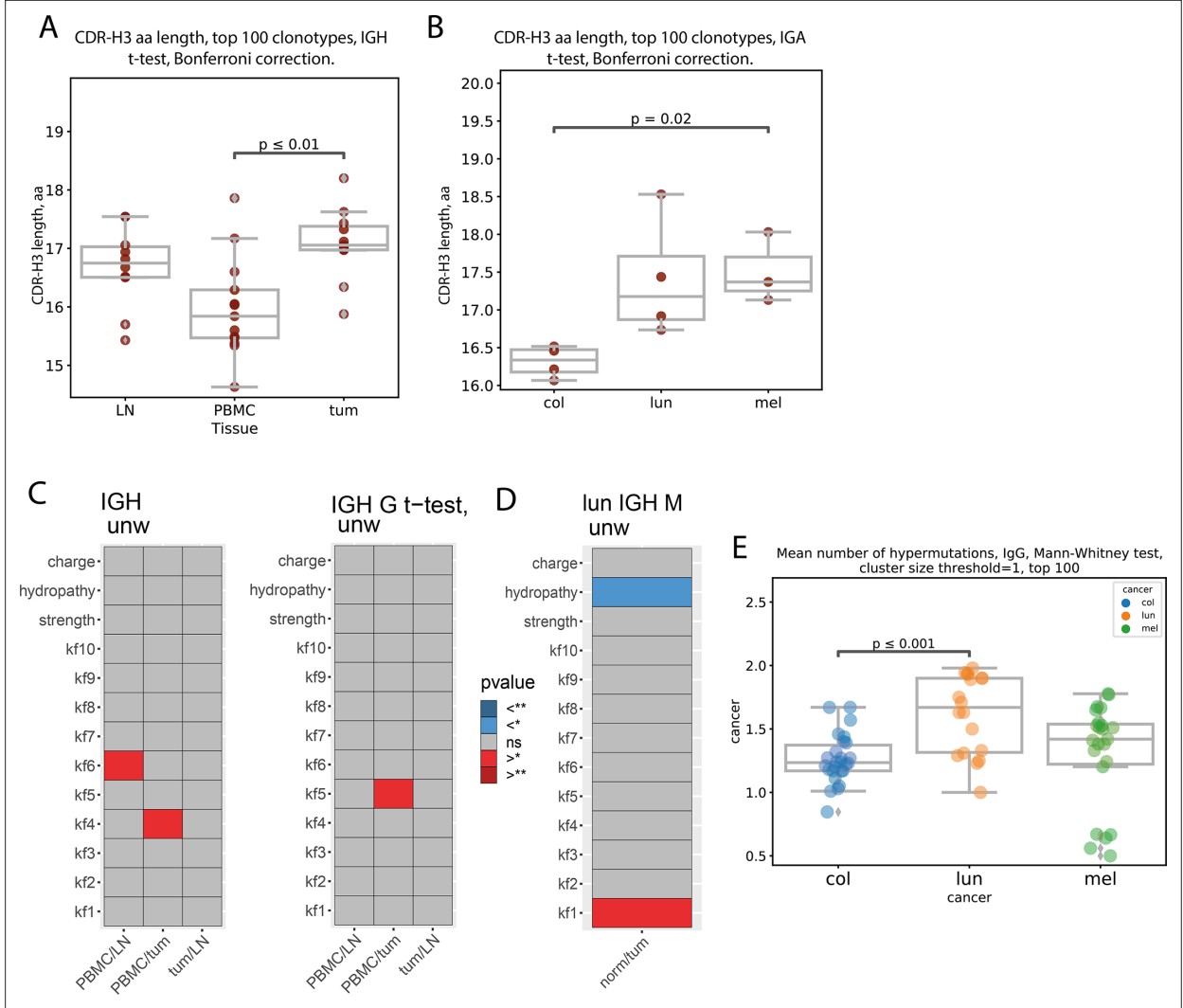

**Figure 6.** LN-to-LN difference of BCR repertoires in colorectal cancer (clonotype level - panels **A**, **B**; clonal group level - panels **C**, **D**; individual patient data). (**A**) Repertoire overlap between tumor and three separate LNs from the draining lymph node pool; Mann-Whitney test, Bonferroni correction. (**B**) Network representation of Ig repertoires from tumor and three separate LNs, with circles representing individual CDR-H3 sequences, size of the circles corresponding to the clone frequency, and color corresponding to the isotype; (**C**) triangle plot visualization of clonal group distribution between three different LNs, with size corresponding to the number of individual CDR-H3 sequences (clonotypes) within a given clonal group, and color corresponding to the percentage of tumor-derived clonotypes–within the clonal group; (**D**) example of a clonal lineage consisting of CDR-H3 sequences derived from a lymph node (blue) and all three tumor fragments (shades of red) from patient ccp2, shapes representing isotypes and size representing frequency of a given sequence in a given sample.

## LN-to-LN heterogeneity in colorectal cancer

Next, we sought to investigate whether within a group of tumor-draining LNs BCR repertoire analysis could discern LNs that were in more intensive clonal exchange with the tumor. This question may be addressed at the individual clonotype level and at the level of clonal groups.

At the individual clonotype level, we compared F2 metric values for pairwise tumor/LN BCR repertoire overlaps from 2 colorectal cancer patients, for which we obtained three separate LNs from the excised surgical material. For patient **ccp2**, a significantly higher overlap of tumor repertoire with one of the LNs was observed (*Figure 6A*). Similarly, Cytoscape network analysis showed more clonotype sharing between LN3 than between other LNs (*Figure 6B*). For **ccp3**, no significant difference was observed between LNs (*Figure 6A*, *right panel*). Similarly, unequal interaction of tumors with LNs was observed at the level of hypermutating clonal groups. We used the proportions of clonotypes originating from LN1, 2 and 3 among all LN-clonotypes within a clonal group as coordinates for triangle

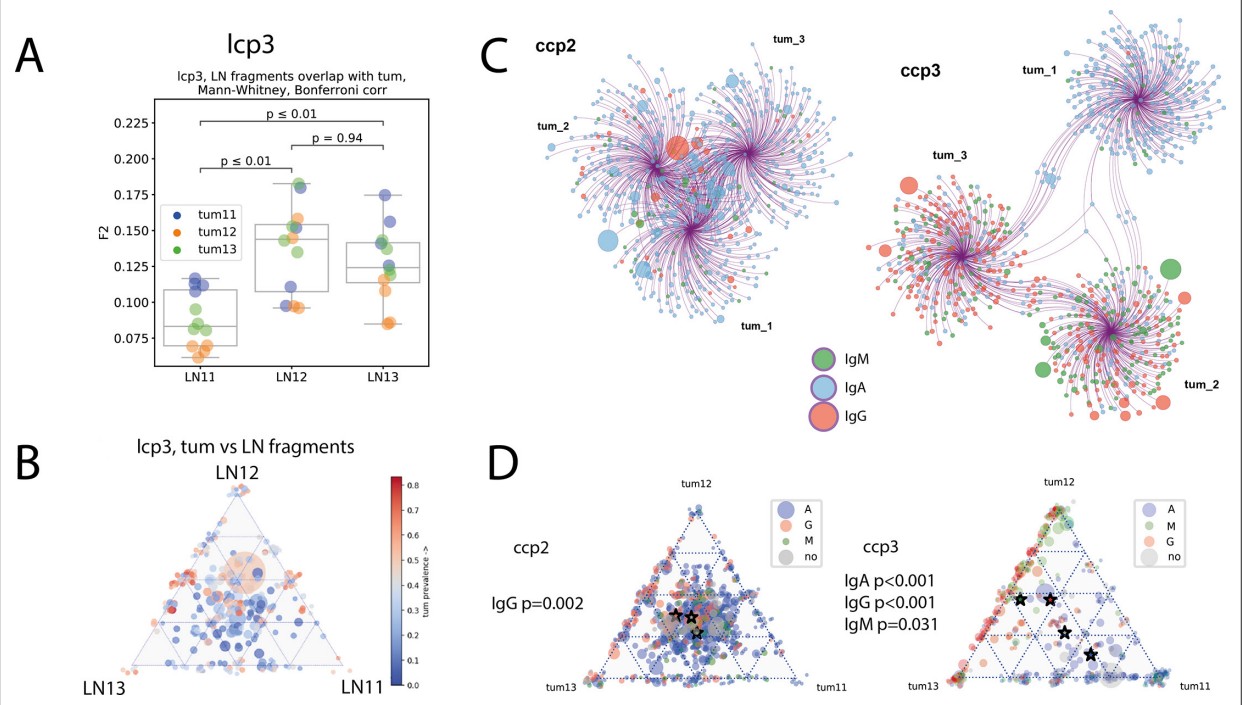

**Figure 7.** Intra-LN and intratumoral heterogeneity (clonotype level - panels **A**, **C**; clonal group level - panel **B**, **D**; individual patient data). (**A**) Repertoire overlap between tumor and lymph node fragments for colorectal cancer patient ccp6 and lung cancer patient lcp3; Mann-Whitney test, Bonferroni corr. (**B**) Triangle plot visualization of clonal group distribution between lymph node fragments, with size corresponding to the number of individual CDR-H3 sequences (clonotypes) within a given clonal group, and color corresponding to the percentage of tumor-derived clonotypes within the clonal group; (**C**) network representation of Ig repertoires from tumor fragments, with circles representing individual CDR-H3 sequences, size of the circles corresponding to the clone frequency, and color corresponding to the isotype, edges connect clonotypes to their fragment of origin; (**D**) triangle plot visualization of clonal group distribution between tumor fragments, with size corresponding to the number of individual CDR-H3 sequences (clonotypes) within a given clonal group, and color representing dominant clonotype; Stars represent non-weighted by size mean center of triangle coordinates. Chi-2 test for goodness of fit is used to test if each tumor segment equally contributes to clonal groups formation.

The online version of this article includes the following figure supplement(s) for figure 7:

**Figure supplement 1.** BCR repertoire overlap for separate fragments of tumor (different), and from cell suspension level replicates from the same fragment (same), patients ccp2 (**A**), ccp3 (**B**), ccp5 (**C**), ccp6 (**D**).

**Figure supplement 2.** Histochemical analysis of lymphoid infiltration in ccp2 sample.

plots (*Figure 6C*). For **ccp2**, some clonal groups were focussed in LN3 and were also prominently represented in the tumor (*red*). However, most of the other clonal groups resided in the center of the triangle, so the center of mass was not shifted towards any of the LNs overall. Functionally, this may again indicate that within a group of LNs, there is some inequality in terms of access to tumor antigens, and this inequality shapes the BCR repertoires within these LNs. On *Figure 6D* we show an example of a clonal group that includes clonotypes from tumor (multiple tumor fragments) and only from one of the LNs, to illustrate asymmetric involvement of LNs in the development of a hypermutating clonal group. To validate this hypothesis, it would be beneficial to obtain BCR repertoires from non-tumor-draining LNs; however, this was not possible in the current study.

These observations reflect a complex interplay between tumors and LNs, which may be involved in the adaptive immune response to tumor antigens.

## Intra-LN heterogeneity

Likewise, we asked whether parts of LNs are equally involved in interaction with tumor. For lcp3, we obtained BCR repertoires from 3 fragments of one draining LN and compared them to tumor repertoires on the individual clonotype and clonal group level. Repertoire from fragment LN11 showed significantly lower overlap with the tumor than other two LN fragments, LN12 and LN13 (*Figure 7A*,

*left*). Triangle plot analysis of clonal group distribution also showed that the majority of the tumor-dominated (red) clonal groups resided in the LN12-LN13 side of the triangle (*Figure 7B*, *left*).

## Intratumoral heterogeneity of immune repertoires

Statistically, the magnitude of intratumoral genetic heterogeneity correlates with the heterogeneity of immune cell infiltration, implying the co-evolution of the tumor genetic architecture and immune microenvironment *Jia et al., 2018*. Spatial heterogeneity of the T cell receptor repertoire is also known to reflect the spatial distribution of mutations in the tumor (*Reuben et al., 2017*; *Joshi et al., 2019*). Furthermore, this is clinically relevant, because higher heterogeneity of tumor-infiltrating T-cell repertoire is associated with higher risk of recurrence and shorter disease-free survival (*Reuben et al., 2017*).

B-cell repertoire heterogeneity, however, remains relatively understudied. The mechanisms underlying tumor infiltrating B-cell heterogeneity may involve not only differential infiltration and accumulation of lymphocytes, but also local development of tertiary lymphoid structures which then drive the development of local immune response specific to subclonal neoantigens.

Here, we aim to quantify intratumoral BCR repertoire heterogeneity, measured as the extent of clonotype or clonal group sharing between parts of the tumor. First, we confirmed for all studied tumor samples, that repertoires derived from separate tumor fragments are significantly less overlapping (F2 metric) than repertoires from biological replicates of one tumor fragment (**Fig. S3**). Then we pooled repertoires from replicates belonging to the same tumor fragments and analyzed the overlap between top 300 clonotypes from fragments of the same tumor using Cytoscape platform (*Figure 7C*). We observed significant patient-to-patient variability in the degree of heterogeneity between tumor fragments. In a tumor from patient **ccp2**, all three fragments had a high number of clonotypes in common (represented also by close distance between anchor nodes on the bubble diagram), and the isotype composition was also similar and dominated by IgA (*Figure 7C and D left*). Overall, this drew a picture of a relatively homogenous immune infiltrate in the tumor of this patient. In patient **ccp3** (*Figure 7C and D*, *right*), we saw a remarkably different pattern. In ccp3, fragment tum_1 was dominated by IgA and had many shared IgA clonotypes with fragment tum_3. Fragment tum_2 shared many clonotypes with tum_3, and these clonotypes were mostly IgG and IgM clonotypes. Heterogeneity of clonal group distribution was also significantly more prominent in **ccp3** compared to **ccp2** (*Figure 7D*). In **ccp2**, significant eccentricity was observed only for IgG-dominated clonal groups (red star on *Figure 7D*, *left*), which tended to share between fragments tum_2 and tum_3, and not tum_1. This may indicate that IgA clonal groups are specific to ubiquitous tumor or self-antigens, whereas IgG clonal groups more likely recognize tumor specific and/or subclonal antigens (present only in some parts of tumor). In **ccp3**, most clonal groups showed a significant degree of eccentricity on triangle plot (*Figure 7D*, *right*), indicating heterogeneous distribution between tumor fragments, for IgA-dominated as well as for IgG-dominated clonal groups. Given that clonality of **ccp3** tumor fragments tum_2 and tum_3 was also higher than that of **ccp2** tumor, we hypothesized that heterogeneity plus high clonality indicate formation of TLSs and thus efficient local immune response towards tumor-related antigens.

To analyze the presence of TLSss in our tumor samples, we used histology and immunohistochemistry. We found frequent dense leukocytic accumulations in the peritumoral region, as well as distant from the tumor. The latter were the mucosal/submucosal layers, whereas peritumoral ones were found in all the intestinal layers (**Fig. S4B**). Such an appearance in the outer layers and prominent accumulation near the tumor indicate that these are more likely to be TLSs, rather than Peyer's patch-like structures that are localized predominantly in the mucosal and submucosal layers. In order to evaluate whether this accumulation has an organized TLSs-related lymphocyte distribution, we used multicolor immunohistochemical staining of T and B cells together with TLSs-related markers: high endothelial venule marker PNAd and CXCL13 chemokine, which drives TLSs formation. We found that CD20 B cells and CD3 T cells were the major cell types within these patches, indicating their lymphoid origin (**Fig. S4 E,F**). Peritumoral structures have condensed B cells follicles surrounded by T cells and high endothelial vessels (HEV), confirming that these are primary follicular TLSs (*Werner et al., 2021*; *Wang et al., 2022b*). Distant lymphoid structures in the mucosal layer have a more dispersed and intermixed T and B-cell distribution, indicating a lower degree of maturation. Similar lymphoid aggregates in the mucosa and peritumoral regions have been observed on all tissue blockers for patients

with colorectal cancer (two for each patient). However, their comparison is unrepresentative, owing to variations in the sampling of tissue fragments.

The design of our study included obtaining cellular-level replicates for each processed tissue fragment. This allowed us to reliably detect CDR-H3 clonotypes that were significantly expanded in a given sample relative to other samples. *Figure 8* shows an example of such analysis. We used the EdgeR library from the Bioconductor package to detect clonotypes that were differentially expanded in separate fragments of tumors. These clonotypes are represented on *Figure 8A and B* as colored circles on frequency correlation plots. Expanded clonotypes with the highest frequencies are also labeled on Cytoscape plots (*Figure 8C*, sequence labels). Interestingly, the most expanded clonotypes were not attributed to any clonal group (and thus were presumably not actively hypermutating), and the largest of these were of IgM isotype. Only one of the expanded clonotypes in this tumor was involved in the hypermutation process, and the structure of its clonal lineage is shown in *Figure 8D*. Conversely, none of the largest clonal lineages detected in this patient contained any expanded clonotypes (*Figure 8E*). In addition, on average, expanded clonotypes had fewer mutations than non-expanded clonotypes, both in tumors and in tumor-draining LNs (*Figure 8F*, **all patients**).

## Short vs long trees - phylogeny analysis

CDR-H3 is the most variable component of the BCR and is also the most important in terms of antigen recognition. However, missing mutations in other segments may lead to difficulties or a complete inability to accurately recover the phylogeny of hypermutations. Moreover, shorter sequencing reads, which may be sufficient to obtain CDR-H3 data, cover the V segment to a lesser extent, which may lead to less accurate germline gene assignment and, consequently, erroneous rooting of the phylogenetic tree.

To estimate the extent to which phylogenetic trees built on short-read data and CDR-H3 sequences alone reflected the 'true' phylogeny recovered from full-length (CDR1 to FR4 regions) sequences, we used two types of BCR libraries obtained from the same lung cancer patient material. Short CDR-H3 sequences were obtained from libraries prepared according to the 5'RACE protocol and sequenced with a 150+150 bp read length. Long sequences were obtained from libraries prepared according to the IgMultiplex protocol and sequenced at 250+250 bp read length. Then, for each dataset, we independently inferred clonal lineages and built rooted phylogenetic trees for each clonal lineage of size five or more (see Materials and methods for more details). To compare the structures of these tree sets, we selected clonotypes with the CDR-H3 sequence and the corresponding V and J genes present in both the long and short trees. We then calculated their average ranks by distance to the root within their clonal lineages and compared the ranks (**Fig. S5A**). The correlation was around 0.7 ($p<6*10^{-16}$). Therefore, phylogeny inferred based on short CDR-H3 sequences generally reflects the 'true' phylogeny inferred from the CDR1-FR4 repertoires. However, the average distance to the root for IgM and IgA/IgG isotypes, which have significant differences for long repertoires, did not show significance for short (CDR-H3-based) repertoires (**Fig. S5B**).

One potential explanation for this could be that a larger sample size and a higher sequencing depth are required for short sequences to yield statistically significant differences. Indeed, it is impossible to avoid sporadic misplacements of short sequences in the tree when the information on hypermutations is limited. On the other hand, we clearly observed a correlation between long and short tree phylogenies. Therefore, we suppose that the reason for this contradiction is an insufficient sample size. However, there is evidence that CDR-H3 data do not always perfectly represent the 'picture' and should be treated with particular attention.

*Grimsholm et al., 2020* showed that memory B cells have, on average, higher Kidera factor 4 values and lower predicted interaction strength than naive B-cells. We assumed that these features may evolve during the affinity maturation process. Therefore, we checked how the mean kf4, strength, and charge of the five central CDR-H3 amino acids depend on the clonotype's distance to the MRCA on the phylogenetic tree. To ensure that there was no bias in phylogeny as a result of short sequencing length and small sample size, we used full-length sequence trees inferred from lung cancer patient lcp3 and full-length repertoire with very high sequencing depth from a healthy donor. However, we did not find any correlation between strength, charge, and **kf4** and the clonotype's position on a tree (**Fig. S5C**). One exception was a very small increase in charge down the tree found in the healthy donor repertoire ($r=0.04$, $p=4.9*10^{-05}$), which, despite being statistically significant, is probably not

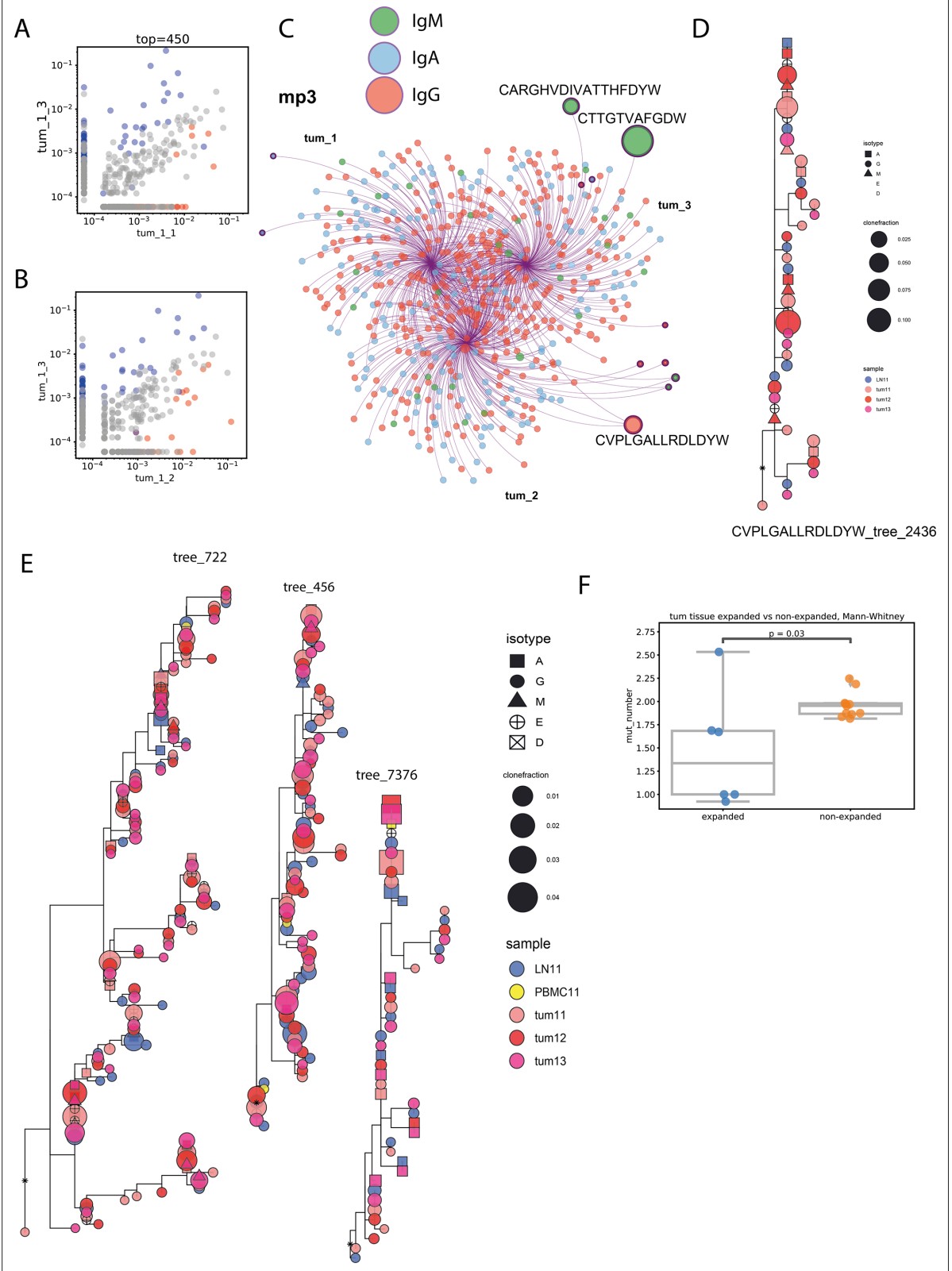

**Figure 8.** Expanded clonotypes and hypermutation analysis (clonotype level - **A**, **B**, **C**, **F**; clonotype group level - **D**, **E**; individual patient data).
(**A, B**) Visualization of expanded clonotypes on frequency correlation plots for pairs of tumor fragments of melanoma tumor from patient mp3;
(**C**) Cytoscape network visualization of top 300 most frequent individual Ig CDR-H3 clonotypes, colored by isotype, with size of circles proportional to the frequency of a given clonotype; (**D**) example of a clonal lineage containing expanded CDR-H3 sequence; (**E**) examples of the biggest clonal lineages,

*Figure 8 continued on next page*

*Figure 8 continued*

none of which contain expanded CDR-H3 sequences; (**F**) average number of mutations in expanded and non-expanded clonotypes; Mann-Whitney test; N=11.

The online version of this article includes the following figure supplement(s) for figure 8:

**Figure supplement 1.** Short (CDR-H3-based) versus full-length IG trees phylogeny analysis.

**Figure supplement 2.** Amino acid properties depending on clonal status Kf4, strength and charge for central 5 aa of CDR3 region in repertoires from PBMC, LN and tumor, for CDR3 clonotypes that belong to a clonal group (has_tree) and those that do not belong to any clonal group (single).

---

mechanistically meaningful. In addition, the largest (>20 members) trees had dN/dS values <1, indicating negative selection pressure (**Fig. S5D**). Taken together, these data suggest that, in the absence of a defined time-controlled antigenic stimulus, such as vaccination, in steady-state equilibrium, only a few immunoglobulin clonal groups show definitive signs of positive selection.

## Productive involvement in hypermutating lineages depends on CDR-H3 characteristics

Taking into consideration the previously found differences between memory and naïve cell features by *Grimsholm et al., 2020*, we hypothesized that amino acid features of CDR-H3 may be selected even before the start of hypermutation and affinity maturation. To test this hypothesis, we checked how the mean kf4, strength, and charge of five central CDR-H3 amino acids depend on the clonal status of the clonotype (**Fig. S6**) and found the results to be tissue-specific. In particular, clonotypes not assigned to any clonal lineage (singles) had lower kf4 values, and hence higher hydrophobicity, than members of clonal groups in the tumor and PBMC repertoires. The predicted interaction strength was higher for singles in PBMC and LNs, but not in the tumor, potentially indicating that tumor-resident clonotypes have undergone selection for CDR-H3 properties, irrespective of their involvement in SHM. Finally, clonal lineages had a higher charge in tumor repertoires, but not in the LNs or PBMCs, indicating a tendency towards polyreactivity in clonotypes evolving locally under the influence of TME.

These results generally match the findings of Grimsholm et al. and support the hypothesis that selection of CDR-H3 amino acid features occurs before cells undergo SHM. Tissue differences could potentially be explained by differences in the proportions of B cell subsets in different tissues.

## Discussion

We performed a multi-localization study of BCR repertoires in cancer patients using an unbiased BCR library preparation technique complemented by deep repertoire sequencing. We analyzed these repertoires from two complementary perspectives. At the level of individual BCR clonotypes, we studied various repertoire features and their differences between tissues of origin, as well as repertoire heterogeneity within tumor or lymph node tissue. At the level of B-cell clonal lineages, we tracked lineage sharing between tissues, and examined differences in clonotype features between clonotypes belonging to any clonal lineage, and those that do not.

First, we compared BCR repertoires from PBMCs and bulk tumor infiltrates and the corresponding sorted CD19$^+$CD20$^-$CD38$^{high}$ plasma cells, and confirmed that the bulk RNA-based BCR repertoire is dominated by plasma cells across different peripheral compartments. This finding is in line with our expectations, given that the IG expression level is 50–500 times higher in plasma cells than in memory B-cells (*Turchaninova et al., 2016*). However, we observed a higher proportion of IgM in PCs within the tumor and a lower proportion of IgG in PCs within the lymph nodes. Higher proportion of IgM in PCs relative to the bulk repertoire may indicate that some IgM-producing plasma cells, while having an appropriate plasmablast/PC surface marker phenotype, produce relatively low levels of antibodies, which leads to them being masked in the bulk repertoire by non-IgM non-PC B cells. A higher level of antibody production correlates with the differentiation of plasmablasts/short-lived plasma cells into long-lived plasma cells. Therefore, one could hypothesize that tumor-infiltrating IgM-expressing antibody-producing cells are skewed towards a less differentiated B-cell compartment compared to other isotypes. Similar logic applies to the IgG-producing plasma cells in the LNs: a lower proportion of IgG-clonotypes in PCs compared to the bulk repertoire indicates that some of the most frequent

IgG clonotypes in the bulk repertoire belong to cells with a non-PC/plasmablast phenotype, presumably stimulated memory B-cells.

The overlap in BCR repertoires and the correlation of isotype proportions between tumor and lymph node samples were higher than those between tumor and peripheral blood samples.A higher repertoire overlap indicates a higher proportion of tumor-related BCR clonotypes in the draining LNs than in the peripheral blood. This underscores the crucial role of draining LNs in the development of the anti-tumor immune response. In addition, consistent with previous studies, our data confirmed that the tumor BCR repertoire is more clonal than that of LNs or PBMCs. We also showed that LNs within the same draining LN pool may be differentially involved in tumor interaction. Overall, LNs are a better source of material to study anti-tumor B-cell response than PBMC, and a better source of material for a potential cancer B-cell or antibody therapy, if tumor tissue is unavailable.

The mean CDR-H3 length of the most frequent BCR clonotypes in tumors was higher than in other tissues. This finding may indicate that tumors tend to be infiltrated with less mature B-cells compared to other compartments (*Grimsholm et al., 2020*). It may also suggest a higher proportion of autoreactive and polyreactive (*Prigent, 2016*; *Boughter et al., 2020*) antibodies, consistent with previous research. However, in colorectal cancer, the mean CDR-H3 length in the IgA repertoire was lower than that in melanoma. This may indicate that the IgA repertoire in colorectal cancer is less influenced by the TME than in other cancers that we studied. Differences in other CDR-H3 properties, such as mean kf3 (b-sheet preference), kf4 (inversely correlated with hydrophobicity), predicted interaction strength, and charge between different tissues, including increased central-CDR-H3 charge in the tumor compared to normal tissue, may suggest a less mature and less specific BCR repertoire of tumor-infiltrating B-cells.

Clonal groups representing hypermutating B-cell lineages differ in their isotype and tissue composition. In colorectal cancer, clonal groups dominated by IgA (>60% of IgA clonotypes) tend to be shared between tumors and blood, whereas IgG-dominated (>60% of IgG clonotypes) clonal groups tend to be shared between tumors and draining LNs. The probability of detection of a given clone in a given location is dependent on the BCR expression level, as well as on the time that these cells spend in a given location. Assuming that BCR is expressed at the same level in all members of a certain clonal population, the proportion of clones within a clonal group derived from a certain tissue is then a snapshot of their preference for the particular tissue of residence or circulation. By comparing IgA- and IgG-dominated clonal groups, we observed subtle differences in the behavior of B cells during ongoing clonal selection and interaction with the tumor. Whereas IgG clonal groups preferentially reside in tumors and draining LNs, IgA-dominated clonal groups show a greater preference for circulation. An obvious question is whether this corresponds to the proportion of isotypes at the cellular level (by surface or intracellular staining). The relative frequencies of antibody-secreting B cells in peripheral blood were IgA1 >IgG1>IgA2>IgM > IgG4 >IgG2>IgG3 (*Lee et al., 1991*), which corresponds to our observations of the peripheral blood bulk repertoire (total IgA >total IgG >total IgM). Therefore, the observed clonal group sharing between tissues may simply be due to the higher number of clonotypes, with a certain isotype detected in a certain tissue. However, for tissue distribution triangle plots, the data were normalized for isotype usage between tissues (i.e. equal numbers of IgG/IgA/IgM most frequent clonotypes were used for PBMC, LN, and blood) to correct for quantitative bias. Therefore, the trend for tissue preference reflects true clonal behavior and not simply quantitative prevalence.

It is well known that only draining LNs show signs of interaction with tumor antigens (*Marzo et al., 1999*). In murine models, non-draining usually occurs contralateral to the tumor site, which is anatomically distant. However, whether LNs within the same anatomical site may differ in terms of their interaction with the tumor has yet to be determined. For some of the patients in this study, we were able to obtain more than one lymph node from the same tumor-draining pool, which allowed us to address this question directly on the level of BCR repertoires. A higher overlap between the tumor and LN repertoire signifies a more intensive interaction with the tumor and its antigens. In at least one patient, the LNs studied were significantly different in this regard, with one of the LNs being significantly more similar to the tumor in its BCR repertoire. This observation was also reproduced at the clonal group level, where the lymph node that was in closer interaction with the tumor in terms of clonal overlap also contained more clonal groups that included tumor-derived clonotypes and were confined to this lymph node only. These clonal groups represent antibody maturation and selection processes involving tumors, and occur preferentially in the LN, which is in tight interaction with the tumor, but

not in the others. Whether this also reflects the response to tumor antigens and the maturation of tumor-specific antibodies remains to be tested.

It has been previously shown that within a single lymph node, individual germinal centers (GCs) share their immunoglobulin clonal composition to some extent, and the offspring of one B-cell clone may be found in several individual germinal centers (*Bende et al., 2007*). During GC reactions, individual GCs become oligoclonal (containing 4–13 major B cell clones with functional *IgVH*, as detected in the pre-NGS era) due to an affinity selection process (*Roers et al., 2000*; *Küppers et al., 1993*; *Jacob et al., 1993*). However, in the case of chronic antigen stimulation, as in cancer, repeated cycles of antigen exposure, memory B-cell reactivation, and germinal center reactions should lead to uniform involvement of the whole LN in interaction with tumors and production of tumor-specific immune responses. Again, we sought to study the spatial heterogeneity of LNs at the level of individual immunoglobulin clonotypes and clonal lineages. We found that the BCR repertoires derived from fragments of a single lymph node were significantly different in their similarity to the pooled tumor BCR repertoire (F2 metric). We conclude that despite chronic antigen exposure, despite the ability of memory B cells to leave and re-enter germinal center reactions and form new germinal centers in subsequent rounds of affinity maturation, the BCR landscape of a lymph node remains significantly heterogeneous in space.

Likewise, for tumor immune infiltrates, it was shown that the TCR repertoire is heterogeneous in space, and this heterogeneity correlates with subclonal mutations within the tumor tissue *Joshi et al., 2019* and likely represents expanded mutation-specific T cell clones. BCRs are expected to be even more clonal, both in LNs and tumors. However, for BCRs, heterogeneity analysis is complicated by the orders-of-magnitude difference in immunoglobulin expression between plasma cells and other B-cells (*Turchaninova et al., 2016*). A rare plasma cell will have a low probability of sampling at the cellular level but will produce a sufficient amount of immunoglobulin RNA to be reliably detected in the BCR repertoire. Therefore, appropriate use of biological (cellular) replicates sequenced at comparable depths is crucial for accurate detection of differentially expanded clonotypes (*Yuzhakova et al., 2020*). Following the logic described above, we were able to analyze repertoire overlap between tumor fragments and detect BCR clonotypes that were differentially expanded in individual fragments. We observed that, in some patients, the most abundant BCR clonotypes were different in different fragments. In this case, the most expanded clonotypes rarely overlapped between fragments, thus probably representing clonotypes that recognize subclonal tumor antigens. This needs to be tested directly and, if proven true, may potentially be a reliable method for the identification of tumor-specific antibodies. In other cases, the tumor may be relatively homogeneous, without significant expansion, and with highly overlapping repertoires of individual fragments. These observations are mostly reproduced at the level of clonal groups, where in heterogeneous tumors, we observed clonal groups that concentrated almost entirely in single tumor fragments, whereas in homogenous tumors, clonal groups mostly include clonotypes derived from all fragments of the tumor.

The physicochemical properties of the CDR-H3 region differ between clonotypes that represent clonal lineages, and thus, are actively hypermutating, and those that do not. The most prominent difference was observed for kf4 (inversely related to hydrophobicity) in PBMC and tumors; clonotypes that undergo hypermutation are, on average, less hydrophobic, which indicates higher binding specificity. However, there was no correlation between kf4 and the extent of hypermutation, even in large full-length Ig-profiling datasets. One possible explanation may be that the selection of less hydrophobic CDR-H3s occurred before the onset of hypermutation.

Interestingly, analysis of silent vs. non-silent hypermutations (dNdS) suggests that in the absence of a defined time-controlled antigenic stimulus, such as vaccination, in steady-state equilibrium, only a few immunoglobulin clonal groups show definitive signs of positive selection. It would be interesting to systematically explore the specificity of these clones in patients with cancer as a potential source of tumor-reactive antibodies.

In general, this study has several limitations. One is the small number of patients, both per cancer type and overall. Another is that our 5'RACE library preparation protocol, which was used for most samples in this study, does not allow for an accurate distinction between the IgA and IgG isotype subtypes. Regarding clonal lineage structure analysis, we found that our study design has a major limitation of a relatively short (150 bp) read length, which does not provide sufficient information for hypermutation analysis.

Nevertheless, our observations contribute to the understanding of the interactions among the tumor immune microenvironment, tumor-draining LNs, and peripheral blood.

## Materials and methods

### Patients

All clinical samples were acquired from the N.N. Blokhin National Medical Research Center of Oncology or Volga District Medical Centre under the Federal Medical and Biological Agency. This study was conducted in accordance with ICH-GCP. The protocol was approved by the Ethical Committees of the Volga District Medical Centre under the Federal Medical and Biological Agency, and by the N.N. Blokhin National Medical Research Center of Oncology. Written informed consent was obtained from all patients.

Fourteen patients were included in the study, of which four were diagnosed with colon cancer, four with lung adenocarcinoma, and six with melanoma. Patients with prior therapy were excluded. For each patient, samples from the tumor (tum), draining LN(s) (further referred to as LN1, LN2, LN3 if multiple LNs were analyzed), peripheral blood (PBMC), and normal tissue (norm) were collected, or only some of these tissues (*Supplementary file 2*). Several distant pieces were resected from each tumor and lymph node, unless there were several LNs available (as for colon cancer patients, ccp2, ccp3). All samples were bulk (unsorted), but for two colon cancer patients (ccp2 and ccp3), plasma cells were sorted from a part of each sample. All the samples were processed and stored for library preparation.

Blood sampling was performed immediately prior to surgery, and the total volume of obtained blood did not exceed 35 ml for each patient. Fragments of tumor, LN, or normal tissue were placed in 50 ml Falcon tubes containing 7–10 ml of MACS Tissue Storage Solution (Miltenyi Biotec, cat. 130-100-008) and stored at room temperature for no longer than 2 hr. All the procedures were performed under sterile conditions. PBMC were isolated from the blood samples using the Ficoll-Hypaque centrifugation protocol. Briefly, whole blood was diluted 2.5 times with sterile PBS buffer, carefully layered over Ficoll-Paque PLUS (GE Healthcare, cat. 17-1440-03) and centrifuged at $600 \times g$ for 20 min. Afterwards, buffy coats were harvested from the interface, washed twice with PBS buffer (20 min $350 \times g$), calculated, and lysed in RLT buffer (QIAGEN, cat. 79216) at 0.5–3 *10^6 cells/sample density.

Tum, LN, and normal tissues were mechanically cut into 3–7 mm fragments and incubated in RPMI 1640 medium (Gibco, cat. 42401042) containing 1 mg/ml Liberase TL Research Grade (Roche, cat. 5401020001) and 30 U/ml DNase I (Qiagen, cat. 15200–50), at $37°\mathrm{C}$ for 30–60 min with gentle shaking. Samples were then processed using a gentleMACS Dissociator (Miltenyi Biotec, cat. 130-093-235) and passed through a 100 um Nylon cell strainer to remove non-dissociated fragments. The resulting cell suspension was concentrated by centrifugation (7 min, 350 g) and lysed in RLT buffer(Qiagen, cat. 79216) 0.1–0.5 *10^6 lymphocytes /sample density. All RLT samples (PBMC, tum, LN, and norm lysates) were stored at –80 °C before preparation of the BCR IGH repertoire libraries.

### PC isolation

For colon cancer patients, three distant samples (at least 3 cm apart) were obtained from the tumor border with a portion of normal tissue. Cells from three digested tumor samples, three LNs (for patients ccp2 and ccp3) and two replicates of PBMC were stained with a panel of fluorescent antibodies: CD45-PerCP/Cy5.5 (BD 564105, clone HI30), CD38-PE (BD 555460, clone HIT2), CD19-Alexa700 (BD 557921, clone HIB19), CD20-BV510 (BD 563067, clone 2H7), CD25-V450 (BD 560458, clone M-T271) and isolated BD FACSAriaIII cell sorter (BD Bioscience). First, lymphocytes were gated as CD45 + cells, and the plasma cell population was isolated as CD20_neg/CD19_low/CD27_high/CD38_high. Sorting was performed using a FACSAria III cell sorter (BD Bioscience) directly into RLT lysis buffer (Qiagen, cat 79216). Cells from the ccp6 LNs were lysed and unsorted.

### BCR library preparation and sequencing

All BCR IGH libraries were generated using the 5RACE methodology described by *Turchaninova et al., 2016* and sequenced with a 150+150 bp read length. For one of the lung cancer patients, lcp3, additional tumor, lymph node, and normal lung tissue libraries were generated using IG RNA

Multiplex kit (MiLaboratories Inc) and sequenced with 250+250 read length. To account for sampling bias, we also obtained technical replicate samples at the cell suspension level.

## Preprocessing of sequencing data

To process sequencing reads, we used the MiNNN software to extract UMIs from raw sequencing reads, correct errors, and assemble consensus sequences for each UMI. For bulk libraries prepared with the 5'RACE protocol and sequenced with 150+150 read length, we filtered out UMI with less than three reads. For bulk libraries prepared with the Multiplex protocol and sequenced with 250+250 read length, we filtered out UMI with less than four reads. For libraries of sorted plasma cells prepared with the 5'RACE protocol and sequenced with 150+150 read length, we filtered out UMI for which there were fewer than two reads.

To ensure the absence of contamination, we checked for the presence of identical UMI in different samples. If such a UMI was identified, and BCR sequences for such UMIs were identical or almost identical (suspecting amplification error), then the number of reads for this UMI in both samples were compared. If the number of reads of this particular UMI in one sample exceeded the number of reads of this UMI in another sample by more than five times, this UMI was eliminated from the sample with a smaller number of reads per UMI. If the ratio was lower than 5, the UMI was eliminated from both samples.

After UMI-based decontamination, we used MIXCR software (MiLaboratories Inc) to assemble reads into quantitated clonotypes, determine germline V, D, and J genes, isotypes, and find the boundaries of target regions, such as CDR-H3. Clonotypes derived from only one UMI were excluded from the analysis of individual clonotype features but were used to analyze clonal lineages and hypermutation phylogeny. Single-UMI sequences were eliminated to avoid errors during cDNA synthesis during library preparation. However, modern reverse transcriptases have high fidelity, with 2.5e-05 error rates for some of them (*Sebastián-Martín et al., 2018*). Therefore, we considered it reasonable to include single UMI sequences in those parts of the analysis, where a larger sample size was important.

Samples with 50 or less clonotypes left after preprocessing were excluded from the analysis.

## Clonal lineage inference

We identified sequences belonging to the same clonal lineage using the ChangeO software. The criteria for the initial grouping were the same V and J germline genes identified by MIXCR, and the same CDR-H3 length. These criteria do not account for the D segment, as there is insufficient confidence in the germline annotations due to its short length and high level of mutations. Sequences within each group were defined as belonging to the same clonal lineage if they had a nucleotide CDR-H3 sequence identity above a certain threshold. Such a threshold was individually defined for each patient's dataset as a local minimum of the distance-to-nearest distribution function (*Gupta et al., 2015*). In most cases, this threshold is set between 80% and 85%.

## Phylogenetic inference

The phylogeny of B-cell hypermutations was inferred for each clonal lineage of size five or more using the maximum likelihood method and the GTR GAMMA nucleotide substitution model. To find the most recent common ancestor (MRCA) or root of the tree, we used an outgroup constructed as a conjugate of germline segments V and J defined by MIXCR. The D segment was excluded from the outgroup formation, as there was insufficient confidence in the germline annotations due to its short length and high level of mutations. The MUSCLE tool was used for multiple sequence alignment and RAxML software was used to build and root phylogenetic trees.

## Sample pooling and normalization

For some parts of the analysis, we considered samples irrespective of technical replicates or specific tumor/LN segments. In such cases, the corresponding repertoire datasets were pooled in one, where for each clonotype, its new frequency was calculated as the mean of the clonotype's frequencies from separate files. Such an approach helps avoid sampling bias and achieve equal contributions of all clonotype sets being pooled.

## Statistical analysis

In our analysis, we often used repertoires of different parts of the same tissue as separate observations within the same comparison. Examples of these could be isotypes, different pieces of tumors

or LNs, or PBMC samples taken at different time points. Understanding the pitfalls of this approach in general, we argue that it can be justified in some cases considering the heterogeneity of tissues, especially tumors, and the distinctive characteristics of different isotypes.

Most of the analysis was performed using VDJtools and custom Python and R scripts.

## Expanded clonotypes detection

We used the EdgeR package in R to identify the clonotypes that were differentially expressed between the two sample sets. The problem with this approach is determining the correct number of counts required to pass into the DGEList function of EdgeR. Using a number of unique UMIs detected for each clonotype in the sample might not be a good idea, considering the possibility of sampling bias (e.g. resecting tumors into pieces of slightly different sizes). To account for sampling bias, we defined clonotype count for the DGEList function as clonotype frequency in a normalized sample multiplied by the total number of unique UMIs in all groups of samples.

The output of the DGEList function is then normalized and passed to the exactTest function of the EdgeR. Clonotypes with FDR <0.05 and logFC >0 were considered expanded in a corresponding group of samples.

## Immunohistochemistry

Sections of formalin-fixed paraffin-embedded tissue were sliced on an RM2235 microtome (Leica Biosystems). Slices were deparaffinized with Xylol and Ethanol. H&E staining was performed using Mayer's hematoxylin and eosin (Biovitrum, Russia). For IHC staining, the slices were demasked in AR9 buffer (PerkinElmer) for 10 min at 98°C in a water bath, washed twice in PBS, and blocked for 30 min with Protein Block (Leica Novocastra, UK). Primary antibodies were added without washing the blocking solution and incubated overnight at 4°C. After incubation with primary antibodies, slices were washed twice in PBS and stained with the NovoLink polymer detection system (Leica Biosystems, UK) according to the manufacturer's instructions or with HRP-streptavidin (Jackson Immunoresearch 016-620-084, 1:500 for 30 min) and contrasted with 10 µM CF tyramide dye (Biotium, USA). The following primary antibodies were used at the consequence and with indicated dilutions and CF dyes: CD20 (Leica Biosystems PA0200), 1:1 with CF488; CXCL13 (GeneTex GTX108471, Taiwan), 1:100 with CF660; CD3 (HuaBio HA720082, China), 1:200 with CF430; PNAd-biotin (BioLegend 120804, USA), 1:200 with CF555. Before staining with non-biotinylated antibodies, the antibodies were stripped in AR9 buffer for 10 min at 98 °C. After staining with antibodies, the slides were counterstained with DAPI (0.2 nM) for 5 min, embedded in Mowiol 4–88 (Sigma-Aldrich), and coverslipped.

Brightfield and fluorescence images were acquired using an EVOS M7000 microscope (Thermo Fisher Scientific) with a 20 x dry objective. The following light cubes were used: DAPI (AMEP4650) for DAPI fluorescence, CFP (AMEP4653) for CF430 fluorescence, YFP (AMEP4654) for CF488 fluorescence, RFP (AMEP4652) for CF555 fluorescence, and Cy5 (AMEP4656) for CF660. Whole slice images were stitched using microscope imaging software. Fluorescence images were contrasted, colorized into pseudocolors, and overlaid using Fiji software (https://imagej.net/).

## Acknowledgements

Supported by grant № 075-15-2019-1789 from the Ministry of Science and Higher Education of the 46Russian Federation.

## Additional information

### Funding

| Funder | Grant reference number | Author |
| --- | --- | --- |
| Ministry of Science and Higher Education of the Russian Federation | 075-15-2019-1789 | Dmitriy M Chudakov |

| Funder | Grant reference number | Author |
| --- | --- | --- |

The funders had no role in study design, data collection and interpretation, or the decision to submit the work for publication.

## Author contributions

Sofia V Krasik, Software, Investigation, Visualization, Writing - original draft; Ekaterina A Bryushkova, Investigation, Methodology, Writing - original draft; George V Sharonov, Investigation, Visualization, Methodology, Writing - original draft; Daria S Myalik, Elizaveta V Shurganova, Dmitry V Komarov, Irina A Shagina, Polina S Shpudeiko, Maria A Turchaninova, Maria T Vakhitova, Investigation; Igor V Samoylenko, Resources, Investigation, Methodology; Dimitr T Marinov, Resources, Methodology; Lev V Demidov, Vladimir E Zagaynov, Resources, Supervision; Dmitriy M Chudakov, Conceptualization, Supervision, Funding acquisition, Methodology, Writing - review and editing; Ekaterina O Serebrovskaya, Supervision, Validation, Investigation, Methodology, Writing - original draft, Writing - review and editing

## Author ORCIDs

George V Sharonov ⓘ https://orcid.org/0000-0001-8610-5054
Dmitriy M Chudakov ⓘ https://orcid.org/0000-0003-0430-790X
Ekaterina O Serebrovskaya ⓘ https://orcid.org/0000-0002-4967-7165

## Ethics

All clinical samples were acquired from the N.N. Blokhin National Medical Research Center of Oncology or Volga District Medical Centre under the Federal Medical and Biological Agency. This study was conducted in accordance with ICH-GCP. The protocols were approved either by the Ethical Committee of the Volga District Medical Centre under the Federal Medical and Biological Agency, or by the N.N. Blokhin National Medical Research Center of Oncology. Written informed consent was obtained from all patients.

Reviewer #3 (Public Review): https://doi.org/10.7554/eLife.89506.4.sa1
Author response https://doi.org/10.7554/eLife.89506.4.sa2

# Additional files

## Supplementary files

Supplementary file 1. Basic characteristics of sequencing output results.

Supplementary file 2. Patient demographics and obtained samples.

MDAR checklist

## Data availability

BCR clonesets are deposited on Figshare under the following link https://doi.org/10.6084/m9.figshare.22340302.v1. Code is deposited on github under the following link https://github.com/vsevolodovna/bcr_in_three_cancers/tree/main (copy archived at *vsevolodovna, 2025*).

The following dataset was generated:

| Author(s) | Year | Dataset title | Dataset URL | Database and Identifier |
| --- | --- | --- | --- | --- |
| Serebrovskaya E, Krasik S, Bryushkova EA | 2024 | Systematic evaluation of intratumoral and peripheral BCR repertoires in three cancers | https://doi.org/10.6084/m9.figshare.22340302.v1 | figshare, 10.6084/m9.figshare.22340302.v1 |

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
