## [Editor Report · eLife Assessment]

This **useful** paper systematically evaluates B-cell receptor (BCR) repertoires across tumors, tumor-draining lymph nodes, and peripheral blood in patients with melanoma, lung adenocarcinoma, and colorectal cancer. It investigates the interplay between the tumor microenvironment and immune responses, revealing differences in BCR clonotype maturity, hypermutation, and spatial distribution. The study highlights the heterogeneity in immune responses and provides **solid** insights into the potential of tumor-infiltrating B cells for therapeutic applications, despite limitations in patient cohort size and sequencing methodology.

---

## [Referee Report · Reviewer #3 (Public Review)]

In multiple cancers, the key roles of B cells are emerging in the tumor microenvironment (TME). The authors of this study appropriately introduce that B cells are relatively under-characterised in the TME and argue correctly that it is not known how the B cell receptor (BCR) repertoires across tumor, lymph node and peripheral blood relate. The authors therefore supply a potentially useful study evaluating the tumor, lymph node and peripheral blood BCR repertoires and site-to-site as well as intra-site relationships. The authors employ sophisticated analysis techniques, although the description of the methods is incomplete.

Major strengths:

(1) The authors provide a unique analysis of BCR repertoires across tumor, dLN, and peripheral blood. The work provides useful insights into inter- and intra-site BCR repertoire heterogeneity. While patient-to-patient variation is expected, the findings with regard to intra-tumor and intra-dLN heterogeneity with the use of fragments from the same tissue are of importance, contribute to the understanding of the TME, and will inform future study design.

(2) A particular strength of the study is the detailed CDR3 physicochemical properties analysis which leads the authors to observations that suggest a less-specific BCR repertoire of TIL-B compared to circulating B cells.

Comments on revisions:

Your efforts in addressing concerns related to methodological details, narrative clarity, and data representation are commendable. The expanded descriptions of Fig. 1A and the experimental design, as well as the restructuring of the discussion, have greatly enhanced the manuscript's clarity and coherence.

---

## [Author Response]

The following is the authors’ response to the previous reviews.

**Reviewer #3:**
Concerns and comments on current version:The revision has improved the manuscript but, in my opinion, remains inadequate. While most of my requested changes have been made, I do not see an expansion of Fig1A legend to incorporate more details about the analysis. Lacking details of methodology was a concern from all reviewers.

To address this concern, we expanded Fig.1A legend, and also significantly expanded the text describing experimental design, to also include the description of the data analysis approach.

“BCR repertoires libraries were obtained using the 5’-RACE (Rapid Amplification of cDNA Ends) protocol as previously described21 and sequenced with 150+150 bp read length. This approach allowed us to achieve high coverage for the obtained libraries (Table S1) to reveal information on clonal composition, CDR-H3 properties, IgM/IgG/IgA isotypes and somatic hypermutation load within CDR-H3. For B cell clonal lineage reconstruction and phylogenetic analysis, however, 150+150 bp read length is suboptimal because it does not cover V-gene region outside CDR-H3, where hypermutations also occur. Therefore, to verify our conclusions based on the data obtained by 150+150 bp sequencing (“short repertoires”), for some of our samples we also generated BCR libraries by IG RNA Multiplex protocol (See Materials and Methods) and sequenced them at 250+250 bp read length (“long repertoires”). Libraries obtained by this protocol cover V gene sequence starting from CDR-H1 and capture most of the hypermutations in the V gene. Conclusions about clonal lineage phylogeny were drawn only when they were corroborated by “long repertoire” analysis.

For BCR repertoire reconstruction from sequencing data, we first performed unique molecular identifier (UMI) extraction and error correction (reads/UMI threshold = 3 for 5`RACE and 4 for IG Multiplex libraries). Then, we used MIXCR58 software to assemble reads into clonotypes, determine germline V, D, and J genes, isotypes, and find the boundaries of target regions, such as CDR-H3. Only

UMI counts, and not read counts, were used for quantitative analysis. Clonotypes derived from only one UMI were excluded from the analysis of individual clonotype features but were used to analyze clonal lineages and hypermutation phylogeny, where sample size was crucial. Samples with 50 or less clonotypes left after preprocessing were excluded from the analysis.”

Similarly, the 'fragmented' narrative was a concern of all reviewers. These matters have not been dealt with adequately enough - there are parts of the manuscript which remain fragmented and confusing.

Unfortunately, the reviewers do not give us a hint as to which parts of the text are the most problematic in their opinion. We identified the parts describing physicochemical properties of CDR3s, Intratumoral heterogeneity and Intra-LN heterogeneity as the most problematic, and edited these parts significantly. Also, we significantly edited the Discussion section (please see the Comparison file for details). Other parts sections were also edited to improve readability and clarity.

The narrative and analysis does not explain how the plasma cell bias has been dealt with adequately and in fact is simply just confusing. There is a paragraph at the beginning of the discussion re the plasma cell bias, which should be re-written to be clearer and moved to have a prominent place early in the results. Why are these results not properly presented? They are key for interpretation of the manuscript. Furthermore, the sorted plasma cell sequencing analysis also has only been performed on two patients.

In response to this concern, we moved the section describing plasma cell bias in the bulk BCR repertoires to the main text.

Another issue is that some disease cohorts are entirely composed of patients with metastasis, some without but metastasis is not mentioned. Metastasis has been shown to impact the immune landscape.

Intrinsic heterogeneity of the cohort is indeed one of the weaknesses of our work, which could negatively impact the statistical significance of our results and, as a consequence, mask certain observations or make them less statistically significant. We mention this in the discussion section. It should not, in our understanding, lead to any false conclusions. We did not, however, pool data from primary and metastatic tumor samples, and all tumor samples that we mention are primary tumors.

The following part of a sentence was added to the discussion:

“...which could negatively impact the statistical significance of our results and, as a consequence, mask certain observations or make them less statistically significant.”

A reviewer brought up a concern about the overlap analysis and I also asked for an explanation on why this F2 metric was chosen. Part of the rebuttal argues that another metric was explored showing similar results, thus the conclusion reached is reasonable. Remarkably, these data are not only omitted from the manuscript, but are not even provided for the reviewers.

We did not intend to conceal any data from the reviewers, and we now added the panel for D metric to the S1 figure. We would also like to point out that the panel describing R metric for repertoire overlaps (a measure of similarity of overlapping clonotype frequencies), was included in the first version of the S2 Figure (now S1 Figure), and it also showed a similar trend. We hope that now the data are fully conclusive.

This manuscript certainly includes some interesting and useful work. Unfortunately, a comprehensive re-write was required to make the work much clearer and easier to understand and this has not been realized.

Again, we thank the reviewers for their thorough evaluation, and hopefully we could make the text clearer in the second reviewed version.